# Thermo-Mechanical Fatigue Crack Growth and Phase Angle Effects in Ti6246

**DOI:** 10.3390/ma15186264

**Published:** 2022-09-09

**Authors:** Jennie Palmer, Jonathan Jones, Mark Whittaker, Steve Williams

**Affiliations:** 1Institute of Structural Materials, Swansea University Bay Campus, Crymlyn Burrows, Swansea SA1 8EN, UK; 2Rolls-Royce plc, P.O. Box 31, Derby DE24 8BJ, UK

**Keywords:** titanium, TMF, transgranular crack growth, thermography, profilometry

## Abstract

A bespoke TMF crack growth test set-up has been developed and validated for use throughout this study and the effects of phasing between mechanical loading and temperature have been investigated. The study shows that TMF cycles may show increased crack growth rate behaviour when compared to isothermal fatigue. The phase angle of the applied TMF cycle can also affect crack growth behaviour, with in-phase (IP) test conditions showing faster crack growth rates than out-of-phase (OP) test conditions. Propagating cracks interact with the microstructure of the material, in particular, the α/β interfaces within the prior beta grains and supporting fractography evidences subtle differences in fracture mechanisms as a result of phase angle.

## 1. Introduction

Titanium and its alloys have seen extensive use in biomedical and aerospace industries, particularly in the gas turbine engine for the past 70 years, due to the unique balance of properties that they offer [1,2]. With their high strength, low density and good fatigue properties titanium alloys have a proven track record of service for front-end components, such as fans and compressors [3]. Only when component temperatures exceed approximately 600 °C are designers compelled to utilise higher density materials, such as nickel-based superalloys since titanium alloys show excessive oxidation at these temperatures.

Whilst the majority of property evaluation for these materials has been conducted under isothermal conditions, this is a simplified and unrealistic version of in-service behaviour. Flight cycles inevitably involve the heating and cooling of components alongside mechanical stressing leading to complex stress-temperature cycles which can only be truly replicated using thermo-mechanical fatigue (TMF) testing. Previous research has shown that the effects of TMF can be more damaging by an order of magnitude when compared to isothermal fatigue at the maximum operating temperature [4]. In order to accurately predict in-service fatigue life, it is critical that the phenomenon of TMF is considered.

TMF testing is experimentally extremely challenging due to the requirements for accurate temperature control but can be carried out under either load or strain control. In either control mode, the interaction of both thermal and mechanical cycles requires definition through a phase angle (ϕ). Considering the case of load controlled TMF, which will be performed in this programme, widely used phase angles include an in-phase (IP) cycle (ϕ = 0°) which can be defined as having temperatures and mechanical stresses increase and decrease concurrently and a fully out-of-phase (OP) cycle (ϕ = 180°), where the temperature decreases as stress increases and vice versa. Clearly, a full range of phase angles between these two extremes is also possible. It should also be noted that cycle direction can be varied, with the test moving either clockwise or anticlockwise through stress-temperature space.

For alloys where lifing approaches are based on damage tolerant approaches, the generation of accurate crack propagation data is critical. The current programme focuses on the titanium alloy Ti-6246 (Ti-6Al-2Sn-4Zr-6Mo). Ti-6246 is widely used for intermediate temperature applications in the gas turbine engine and displays a fine Widmanstatten microstructure, different from many traditional alloys, such as Ti6-4 [5]. Ti6-4 is traditionally replaced by alloys that show improved performance above 350 °C and as such, creep deformation is not recognised as a significant concern. However, in attempting to maximise the usage of titanium alloys in the gas turbine engine, alloys such as Ti-6246 and Ti-834 (Ti-5.8Al-4Sn-3.5Zr-0.7Nb-0.5Mo) are used extensively at higher temperatures [6].

The current paper, therefore, seeks to investigate and understand the behaviour of Ti-6246 under TMF loading conditions. Crack growth behaviour in titanium alloys under TMF loading is an area of research that has received only limited consideration [7]. However, as in-service temperatures rise and TMF cycles become more prevalent, it is critical that this area is understood to ensure that non-conservative growth models are not utilised. The aim of the current work is, therefore, to understand the effects of phase angle and cycle temperature, and how changes within the cycle relate to the interaction between fatigue and higher temperature damage mechanisms such as creep and oxidation. 

## 2. Materials and Methods

### 2.1. Material

Ti6246 rectilinear specimens with nominal dimensions of 7 × 7 mm and a parallel gauge length of 20 mm were utilised for crack propagation testing under mode I loading, where a slot is machined into one corner to initiate crack growth. Crack growth was measured using a well-established Direct Current Potential Drop (DCPD) technique [8] where initial and final crack lengths were determined and compared to voltage readings, with an assumed linear relationship between these two points for a/W values of less than 0.4 (all tests were terminated at this point) [8], as referenced in ASTM-E647 [9]. Stress intensity factors for this specimen were then determined based on the work of Pickard [10]. Crack growth rates are determined using a five-point secant method to provide smoothed data; 0.05 mm diameter platinum wires were welded across a 0.2 mm wide notch at approximately (±0.05) 0.3 mm apart. The slot depth was 0.35 mm. Thermal profiling was conducted prior to TMF crack propagation (TMFCG) testing and isothermal fatigue crack growth (IFCG) tests were conducted in parallel to the TMFCG programme to provide baseline information.

### 2.2. Heating Method

#### 2.2.1. Isothermal Comparison

Conventional radiant furnaces are often considered inappropriate for TMF testing due to the requirement for consistent heat distribution throughout a test sample during the rapid heating profiles used in this type of testing. It is, therefore, important to ensure a viable alternative is used. Induction heating has proven a popular choice for TMF testing, being employed in several institutes, as it is able to provide rapid thermal cycles and can be used for both load and strain-controlled testing [11]. Induction coil systems (ICS) provide heating by passing an alternating current (AC) through a water-cooled copper induction coil to create an electromagnetic field. Current flows against the resistivity of the alloy when a specimen is placed within this electromagnetic field, thus producing eddy currents and resulting in the heating of the specimen. The importance of coil design has been highlighted by Evans et al. [12] and the resultant thermal gradients have been explored by Beck et al. [13]. The same authors highlight concerns when combining induction heating with crack monitoring techniques, such as the direct current potential drop (DCPD) method. This approach is based on a discontinuity (such as the crack in the test piece) being recorded when a homogeneous current of an adequate value is passed through the whole cross-section of the specimen, causing a potential drop; thus it is thought that the eddy currents produced from the induction heating may interact with the current required for DCPD monitoring, causing noise and interference. However, research conducted by Pretty [14] shows the two methods to be a viable combination for TMFCG testing, providing care is taken in selecting an appropriate induction coil design.

The use of a novel quartz lamp furnace set-up was also investigated as part of this research [15,16]. The bespoke radiant lamp furnace (RLF) comprises three independently controllable heating zones and benefits from built-in internal compressed air specimen cooling. Adapted from its original application, this set-up provided another possible method for TMFCG testing. 

Load-controlled IFCG tests were conducted under R = 0 loading conditions at 500 °C with a maximum stress of 500 MPa (trapezoidal 1-1-1-1 waveform) using a conventional radiant furnace, an induction coil system (ICS) and a bespoke radiant lamp furnace (RLF) for the source of heating. These preliminary tests were conducted to provide a comparison between the heating methods and to aid the validation of the ICS and/or the RLF as viable methods of heating during TMF testing. 

#### 2.2.2. TMF Comparison

IP and OP TMFCG tests were carried out using the ICS and RLF using the same loads as the isothermal testing at temperatures up to 550 °C (R = 0, maximum stress of 500 MPa) and temperature ranges from 200 to 500 °C and 200 to 550 °C. A triangular waveform with a frequency of 0.0125 Hz (80 s cycle time) was applied to the tests. As with previous isothermal tests, N-type thermocouple wire of diameter 0.05 mm was welded to the specimen in accordance with BS ISO 12111:2011 [17] in order to measure temperature during the TMF cycles. The temperature was logged using a National Instruments USB-6001, USB multifunction I/O device, capable of logging 20 kS/s and 14-bit resolution. 

### 2.3. Induction Coil Comparison

As previously mentioned, Evans et al. highlighted the importance of coil design, and thus a comparison of different ICSs was conducted [12]. Because of its ability to load/unload the specimen with minimum disruption to the test setup, the originally preferred design is shown in Figure 1a. This design aids test repeatability, and also gives good accessibility to the corner crack, thus making it possible to integrate an infrared camera into the set-up or even adapt the set-up for strain-controlled/strain-monitored testing. However, initial tests showed that although it was possible to achieve a reasonable temperature distribution, it was noted that the transversal field split helical coil appeared to experience interference with the current, resulting in unreliable crack length measurements [15].

Through the use of a non-uniform multi-turn longitudinal field helical coil, as shown in Figure 2a, it was possible to eradicate any current interference whilst also providing improved uniformity in the thermal profile as shown in Figure 2b. As highlighted in previous publications by Palmer et al. this coil design allows for a thermal profile that is consistent with the guidelines outlined in the validated code of practice for stress-controlled TMF testing (gauge length temperature variation < ±5 °C, axial temperature gradients <±10 °C within the gauge length [15,18]). Figure 2b shows that the temperature range achieved using the latter coil is ~6 °C, whereas the former coil design displayed a temperature range of ~12 °C, as shown in Figure 1a.

### 2.4. TMF Temperature Comparisons for Different Heating Methods

As the induction heating process results in heat being generated inside the specimen itself, it is likely the specimen is reasonably uniformly heated throughout during the rapid heating cycle required to represent component loading conditions and to avoid impractically long TMF test durations. However, this is not the case when using an RLF, which relies on heat conduction and thus results in the possibility of thermal gradients during the thermal cycle. To determine such effects, thermal profiling was conducted using both the ICS and RLF. In order to do this, a type-N thermocouple was spot-welded to the centre of each of the four faces of the test piece and one contact type-N thermocouple was placed longitudinally through the specimen to the centre plane by means of a 1 mm hole. 

As anticipated, Figure 3a,b show that during the heating phase in the first 40 s of the cycle, consistent temperatures across all thermocouples are achieved with the ICS; Figure 3b shows that a temperature gradient from the surface to the centre of the specimen exists when using the RLF. During the subsequent cooling phase, the use of external forced air-cooling results in a temperature gradient in both cases, with the centre of the specimen cooling more slowly. This issue is resolvable by increasing the cycle time and thus allowing more time for the specimen to reach a uniform temperature throughout. However, at the critical point of the crack tip, the values in Figure 3 should represent a worst-case scenario as crack growth is limited to a/W < 0.4. 

### 2.5. Localised Crack Tip Heating in ICS

One of the potential difficulties in using induction heating for crack propagation testing is the potential for it to cause localised heating of the crack tip. This was investigated using an Infra-Red Thermal Camera (IRTC) whilst the temperature at a spot-welded type-N thermocouple in the crack plane was controlled to a constant temperature of 500 °C. This measurement was taken slightly ahead of the crack tip to avoid interference from the crack section. Dynamic cycles were also investigated, although the isothermal condition represents the extreme case where power is continually provided through the ICS. By displaying a thermographic image of the test piece, as well as the longitudinal profile of the temperature measured graphically, Figure 4 clearly indicates that there is uniformity in the temperature distribution, indicating no effect of crack tip heating [15].

### 2.6. Final Set-Up

The final set-up used for this TMFCG research includes an ESH (Birmingham, UK) 100 kN servo-hydraulic test machine with a ZwickRoell control cube servo controller and a Cubus software system. The thermal cycle was achieved using an internally water-cooled induction coil for heating and four Meech (Witney, UK) air amplifiers (as used for the profiles provided in Figure 3) in order to provide sufficient, uniform cooling. Using type-N thermocouples, the temperature was controlled and monitored through Dirlik DCPD-TMF software. The final set-up is shown in Figure 5.

For temperature measurement purposes, type-N thermocouples were welded to the specimen surface. In order to ensure that the dominant crack is initiated from the machined notch and not the welded regions, a specific pre-cracking procedure was developed which was shown to minimise this undesirable behaviour. This pre-cracking procedure involved load shedding at room temperature from the original test stress which was higher than the applied stress during the TMFCG test.

### 2.7. Thermal Profile

No code of practice currently exists for TMFCG testing, so temperature control in the current work has been aligned with existing stress-controlled test standards [19]. As described by the standard, the set-up was verified ahead of testing and a thermal profile was recorded by isolating the temperature and recording/monitoring the thermal cycle using type-N thermocouples.

Twelve thermocouples were spot-welded to the gauge of the specimen, as shown in Figure 6b. Thermocouples C1–C4 were located at the centre of each face, along the crack plane. T1–T4 and B1–B4 were located ±2 mm longitudinally from the centre, respectively. Figure 6a displays the thermal profile achieved. Adhering to the validated code of practice for stress-controlled TMF testing, this is the triangular profile used throughout the test programme [20].

## 3. Results and Discussion

### 3.1. Isothermal Comparison

As previously mentioned, the ICS was validated as a viable heating source through preliminary IFCG tests, along with establishing DCPD as a reliable crack monitoring technique. Previous work by Pretty et al. utilised an ICS and adopted DCPD for crack monitoring in the TMFCG testing of nickel alloys, showing that potential interference from the ICS in the DCPD crack monitoring did not significantly affect data collection. The authors were able to conclude that such a set-up is compatible with both IFCG and TMFCG [15]. Similarly, it is important to ensure the bespoke Radiant Lamp Furnace (RLF) based on quartz lamps also produces comparable isothermal results. The data from the preliminary investigation conducted within this research at a cycle temperature of 500 °C are displayed in Figure 7. As can be seen, the measured crack growth rates follow a similar trend and are within a reasonable degree of scatter from each other, showing that comparable results are achieved using all three heating methods.

### 3.2. TMF Comparison

With the baseline isothermal data presented, it is possible to compare the IP and OP TMFCG results using the ICS and RLF, displayed in Figure 8a,b. Under both IP and OP conditions, Figure 8 shows the ICS and RLF produce comparable results and are within reasonable scatter from each other. This also further supports that the temperature gradients shown in Figure 3 do not significantly affect the TMFCG results, although further testing should be considered to evaluate a small systematic increase in growth rate when using the RLF. 

### 3.3. Isothermal vs. TMF

Figure 9 compares the IFCG with the TMFCG results. When interpreting the results, it is important to consider that the IFCG tests are undertaken using a trapezoidal 1-1-1-1 waveform, whereas the TMFCG tests utilise a sinusoidal waveform over a period of 80 s. As shown in Figure 9, at T_MAX_ = 500 °C, the slower TMF cycle is more damaging, presenting a faster crack growth rate. Previous work by Pretty et al. also considered similar effects and concluded that despite the temperature changes in the TMF test, the prolonged cycle results in exposure to high temperatures for an increased time than during the IF test. Due to this, time-dependent damage mechanisms, such as oxidation and creep, significantly influence the material and result in an increased growth rate [14].

However, Figure 9 shows that the crack growth rates are more similar for the 550 °C IFCG and 200–550 °C TMFCG tests than for the 500 °C IFCG and 200–500 °C TMFCG tests. Previous research, carried out by Evans et al. (2005), has shown the influence of oxidation on Ti-6246 at temperatures of 500 °C and above concluding that at these temperatures it becomes the dominant damage mechanism in Ti-6246 [21]. This, therefore, could explain the reason for the lack of effect of phasing in the TMFCG tests where T_MAX_ = 550 °C and that the crack growth rates are more similar for the 550 °C IFCG and 200–550 °C TMFCG tests than the 500 °C IFCG and 200–500 °C TMFCG tests. It is also noticeable that both TMF tests and the 550 °C isothermal test show increased crack growth rates at low values of ΔK, and subsequently lower gradients in the Paris crack growth region than the isothermal test at 500 °C. This is further evidence of the influence of oxidation in these tests where oxide growth ahead of the crack tip dominates the increment of crack growth at lower ΔK values, giving way to a larger contribution from fatigue as the crack length increases. 

### 3.4. Phase Angles

Comparing the IP TMFCG test results to the OP TMFCG test results at 200–500 °C in Figure 10, it is evident that when testing IP, the crack growth rate is faster than when testing OP. This is as expected since, during IP testing, the specimen is exposed to maximum stress (and hence a high value of ΔK) and temperature at the same time, which can be more damaging to the fatigue life. Figure 10 shows that there is a difference in the crack growth rates with varying phase angles, with the crack growth rate slowing as the phase angle approaches the OP condition. However, in comparison to the Paris curves presented for a nickel-based superalloy by Pretty et al., it is seen that the disparity in crack growth rate in relation to phase angle is much larger in the results for the nickel alloy [12,18].

### 3.5. Fractography

Previous research has demonstrated that nickel-based superalloys undergo a change in fracture mechanism with IP test conditions resulting in an intergranular fracture and OP test conditions evidently presenting as a transgranular fracture [18,22]. Initial inspection, on a scanning electron microscope (SEM), shows that independent of the test conditions, Ti-6246 appears to fracture predominantly in a transgranular manner, although there is evidence of mixed mode fracture where cracking interacts with interfacial boundaries between the alpha and beta phases. This is presented in Figure 11. 

Further fractographic analysis of specimens tested under TMFCG conditions shows that subtle differences in the fracture surface are present, with the IP condition presenting as a slightly mixed mode with more rigid, intergranular/interlamellar features from near the pre-crack region (Figure 12a–c), all the way through to near the crack tip. In contrast, the OP condition presented in Figure 12d–f has a much flatter, transgranular appearance across the whole fracture surface. Researchers have reported that creep and oxidation damage ahead of the crack tip could be an attributing factor to the fracture mechanisms seen under IP conditions in nickel-based superalloys [23]. Considering the results in Figure 10, as the phase angle changes from OP to IP, the amount of time per cycle that the crack tip spends at high temperature and tensile stress increases. This is reflected by more evidence of creep and oxidation on the fracture faces and higher crack growth rates.

### 3.6. Profilometry

Profilometry was performed on the fracture surfaces of IP and OP specimens tested at 200–500 °C and 200–550 °C, using a Bruker Alicona InfiniteFocus optical microscope. The surface roughness measurements were calculated in accordance with ASME B46.1-2002, Assessment Surface Topography (Blunt/Jiang 2003), Characterisation of Roughness (Stout 2000) and ISO 25178 Areal–Part 2 [24,25,26,27].

Figure 13a–d show that at both 200–500 °C and 200–550 °C, IP test conditions result in a rougher fracture surface with average surface roughness (S_a_) values of 90.7 µm and 95.2 µm, respectively. This is more than double that of the OP test conditions which have S_a_ values of 42.74 µm and 39.71 µm, respectively, and further supports the findings that there is a difference in fracture mechanism between IP and OP test conditions.

### 3.7. Strain Analysis

Using electron backscatter diffraction (EBSD), strain contouring maps have been plotted to analyse the strain distribution at the crack tip for both IP and OP test pieces at 200–500 °C and 200–550 °C. Colour changes from blue to red indicate increased lattice strain as measured by EBSD. It is evident from Figure 14a–d that at both temperature cycles, the OP test conditions appear to result in the specimen experiencing more strain around the crack tip than the IP test conditions. This again could be attributed to creep damage ahead of the crack tip in the IP tests, causing stresses to be relieved throughout the specimen during testing. Similar to creep damage evidenced in micrographs published by Evans et al., Figure 15a,c show crack branching, tearing and interaction with the lamellae that suggests possible creep damage ahead of the crack tip at distances of up to 10 µm [21]. It appears that the crack branching, in particular, is a result of creep damage along the lamellae interfaces, which the crack interacts with. However, the dominant fatigue crack continues orthogonal to the applied stress. It is clear, though, that at a peak temperature of 550 °C the high stresses at the crack tip combine with this temperature to markedly increase creep damage, producing a more bifurcated crack, Figure 15c. Similar damage is less prevalent in the OP specimens shown in Figure 15b,d, and a sharper crack tip is a result of the crack opening at a lower peak temperature where no creep relaxation and blunting occurs.

## 4. Conclusions

TMFCG testing is paramount to understanding the in-service behaviour of Ti-6246 and enabling accurate lifing models to be developed. The research outlined in this paper has provided a technique that will be crucial to the development of in-service lifing of titanium alloys, in particular, showing that TMF cycles do not significantly increase crack growth rates in this alloy at typical service temperatures. Alongside this finding, the following conclusions can be drawn:Comparing IF tests conducted in the conventional furnace, ICS and RLF, it is evident that all methods result in similar and comparable fatigue crack growth rates. Conducting TMF CG tests in the ICS and RLF also results in similar and comparable fatigue crack growth, although the current work has indicated the potential for slightly increased growth rates when using the RLF. It is also evident, upon comparing the IF and TMF results, that cycle time appears to affect the crack growth rate, with a slower cycle time being more damaging presumably due to due to the longer exposure time at high temperatures.Paris curves show that crack growth dependence on phase angles is more subtle in Ti-6246 than in nickel superalloys, with TMF crack growth rates in the IP test showing the fastest growth rates, and 180° OP the lowest. Fractography shows that the difference in fracture mechanisms, between IP and OP, is very subtle. Both appear to be predominantly transgranular; however, IP shows subtle evidence of more intergranular/interlamellar features. This is unlike what has been reported in nickel superalloys, which demonstrates a clear mechanism change.In the specimens analysed to date, IP fracture surfaces are rougher than OP, with a surface roughness of more than double. This is further supported by microscopy along the fatigue crack which shows crack branching, tearing and evidence of creep damage in IP tests but not in OP. This supports the theory of more environmental interaction in the IP tests.

## Figures and Tables

**Figure 1 materials-15-06264-f001:**
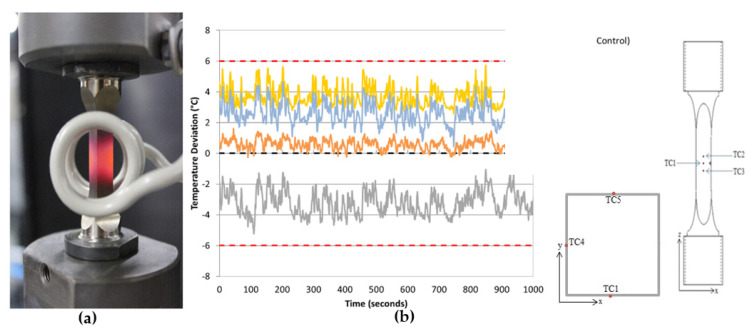
(**a**) Transversal field split helical coil (**b**) Temperature distribution associated with transversal field split helical coil, showing spread of ±6 °C in temperature across the critical volume of material around the crack and TC location, where TC2 and TC3 have been placed at ±5 mm from TC1 respectively, and TC4 and TC5 are placed to show effects at 90° and 180°, respectively [13].

**Figure 2 materials-15-06264-f002:**
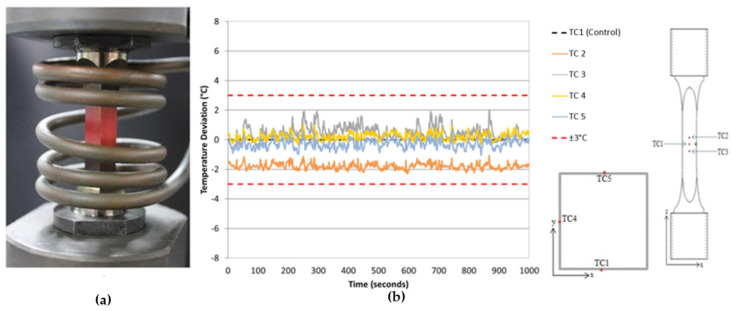
(**a**) Non-uniform multi-turn longitudinal helical coil (**b**) Temperature distribution associated with non-uniform multi-turn longitudinal field helical coil, showing a spread of ±3 °C in temperature across the critical volume of material around the crack and TC location, where TC2 and TC3 have been placed at ±5 mm from TC1, respectively, and TC4 and TC5 are placed to show effects at 90° and 180°, respectively [13].

**Figure 3 materials-15-06264-f003:**
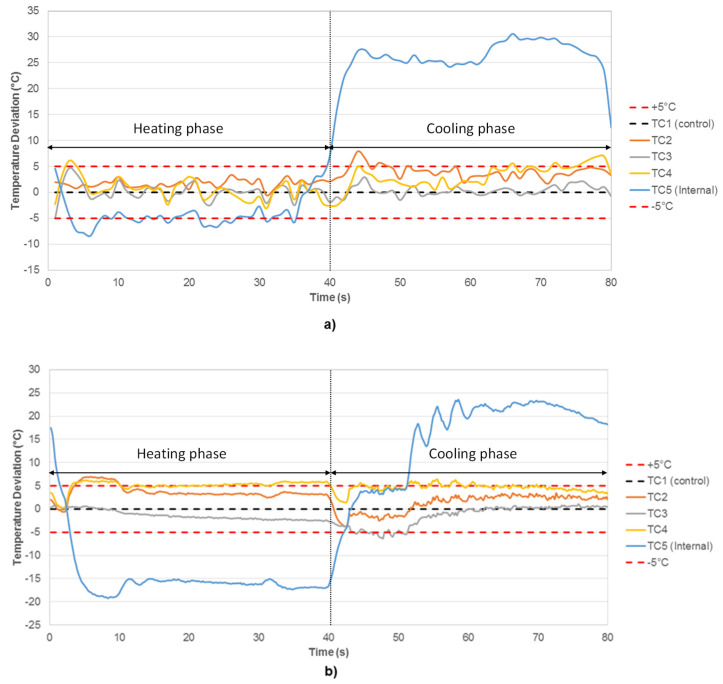
Temperature deviation over the period of an 80 s cycle using (**a**) ICS and (**b**) RLF, showing a temperature gradient on cooling of the specimen in both cases.

**Figure 4 materials-15-06264-f004:**
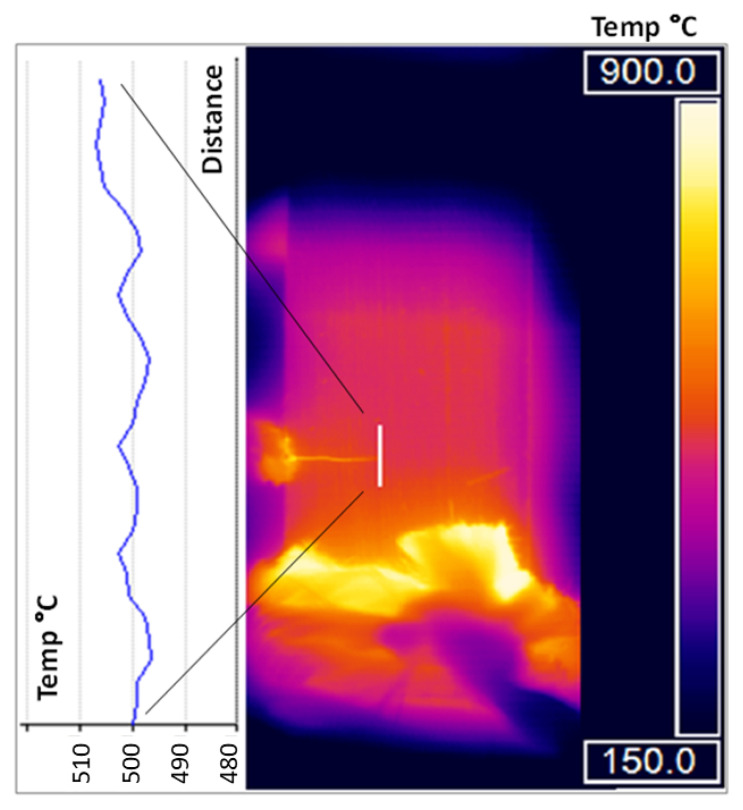
Thermographic image of Ti-6246 with crack plane at 500 °C. Longitudinal profile indicates no effect of crack tip heating.

**Figure 5 materials-15-06264-f005:**
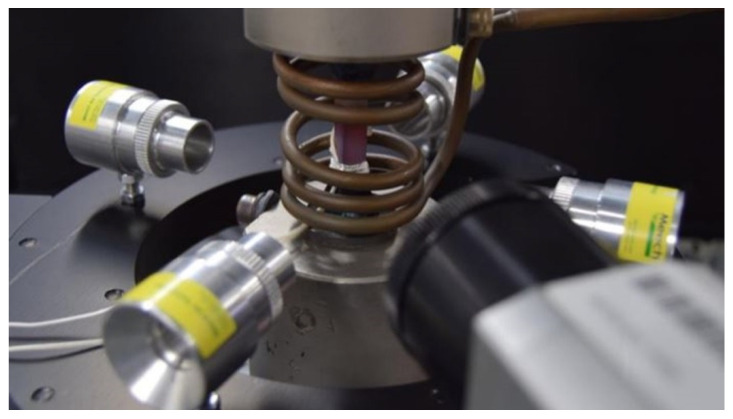
Final setup utilised for TMFCG testing. As shown, a non-uniform multi-turn longitudinal helical field induction coil is adopted for the heating and four air amplifiers, situated on a platform, are used for the cooling. Additionally integrated, is an infra-red camera [15].

**Figure 6 materials-15-06264-f006:**
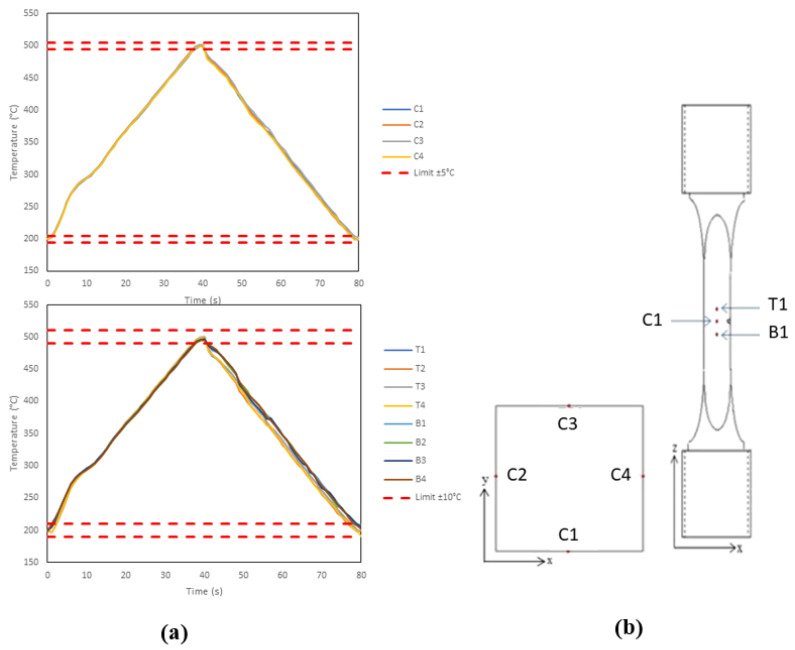
(**a**) Thermal profile achieved for triangular waveform to be used during the test programme and (**b**) thermocouple (TC) location.

**Figure 7 materials-15-06264-f007:**
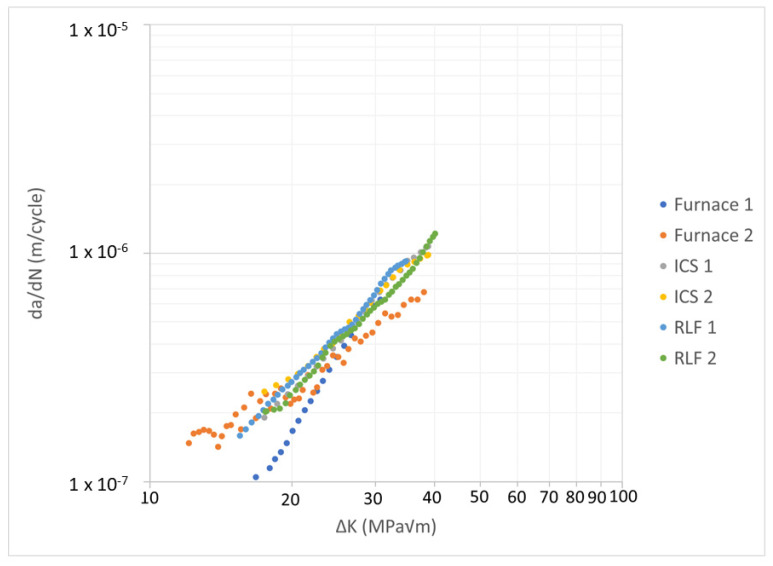
Paris curves comparing isothermal tests conducted using a conventional furnace, ICS and RLF. Tests conducted at 500 °C, 500 MPa, R = 0 and frequency of 0.25 Hz.

**Figure 8 materials-15-06264-f008:**
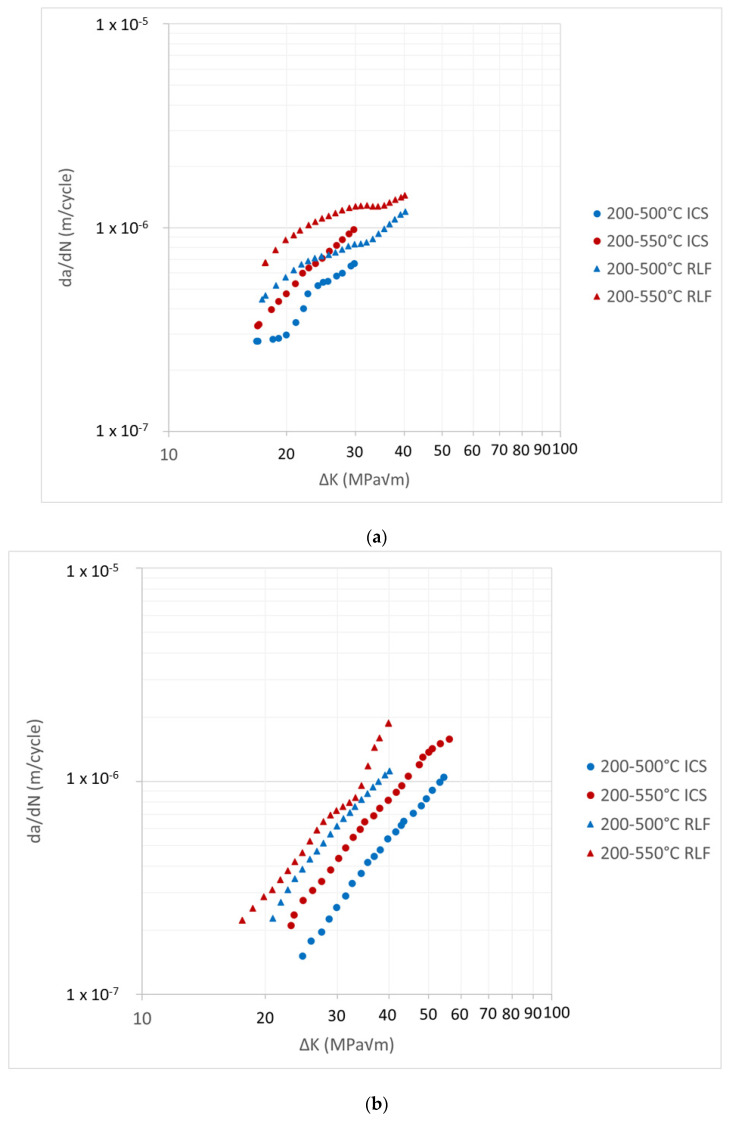
Paris curves comparing TMFCG (**a**) IP and (**b**) OP, using an ICS and RLF. Under both IP and OP conditions, the results are comparable and within reasonable scatter.

**Figure 9 materials-15-06264-f009:**
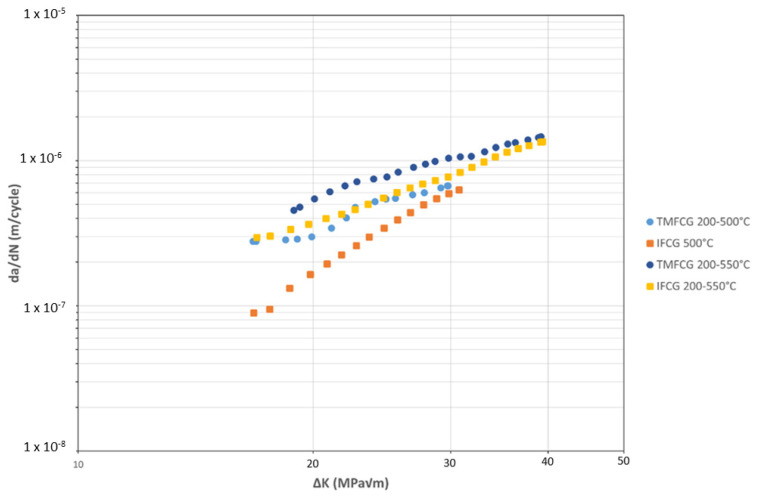
Isothermal fatigue crack growth (IFCG) and IP TMF crack growth (TMFCG) data in Ti6246.

**Figure 10 materials-15-06264-f010:**
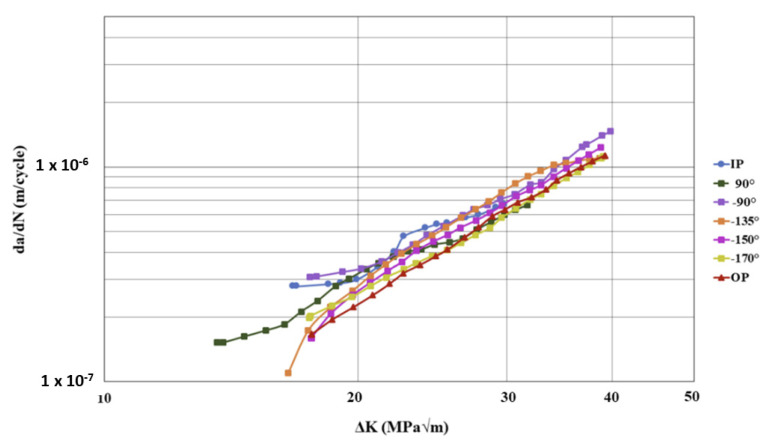
Paris curves comparing the effects of phase angle under TMFCG conditions on Ti-6246 with a temperature cycle of 200–500 °C, maximum stress of 500 MPa and R = 0.

**Figure 11 materials-15-06264-f011:**
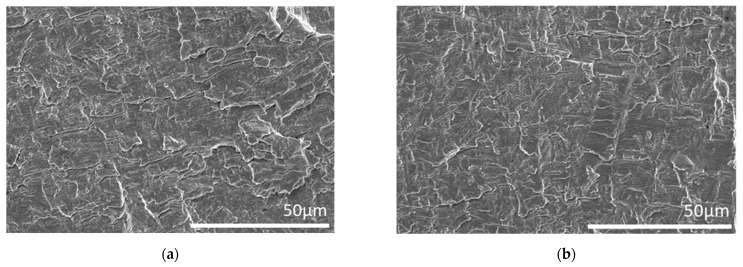
Fractography of (**a**) IFCG 500 °C (**b**) TMFCG IP 200–500 °C and (**c**) TMFCG OP 200–500 °C, all of which show transgranular/mixed-mode cracking.

**Figure 12 materials-15-06264-f012:**
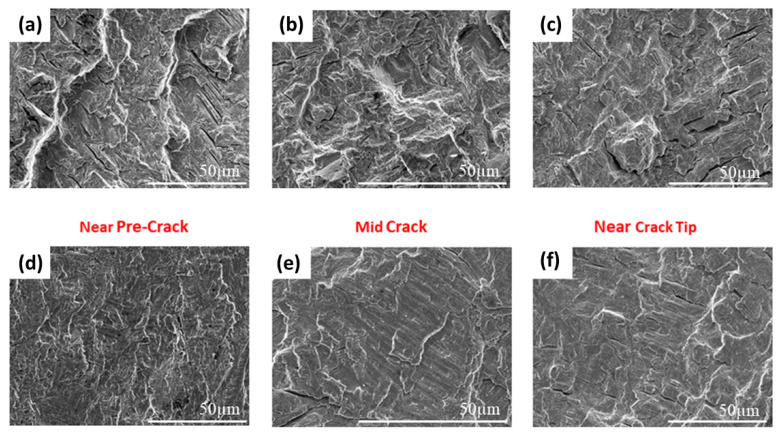
Fractography of TMFCG IP 200–500 °C (**a**) near pre-crack (**b**) mid crack (**c**) near crack tip and TMFCG OP 200–500 °C (**d**) near pre-crack (**e**) mid crack and (**f**) near crack tip.

**Figure 13 materials-15-06264-f013:**
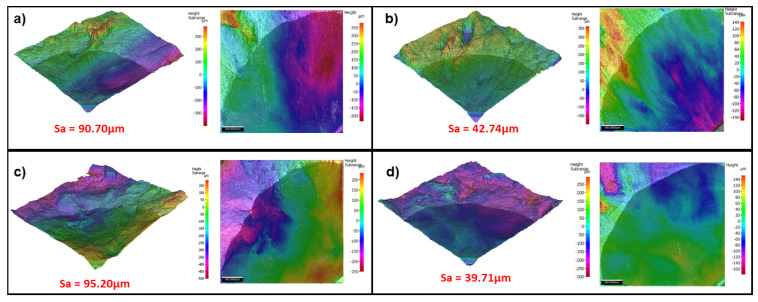
Profilometry of TMFCG fracture surfaces: (**a**) IP 200−500 °C (**b**) OP 200−500 °C (**c**) IP 200−550 °C and (**d**) OP 200−550 °C.

**Figure 14 materials-15-06264-f014:**
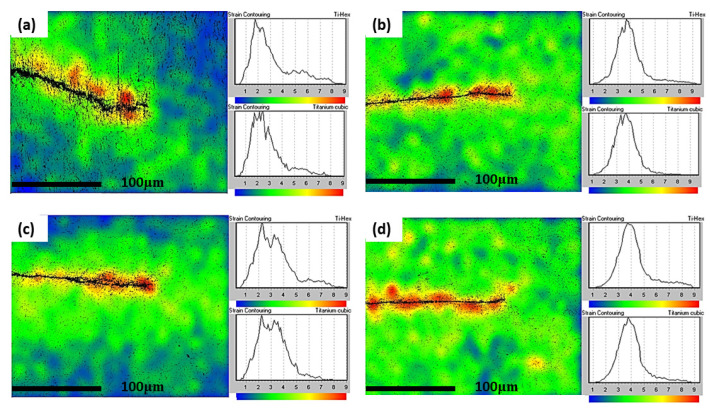
Strain contouring maps (**a**) IP 200−500 °C (**b**) OP 200−500 °C (**c**) IP 200−550 °C (**d**) OP 200−550 °C.

**Figure 15 materials-15-06264-f015:**
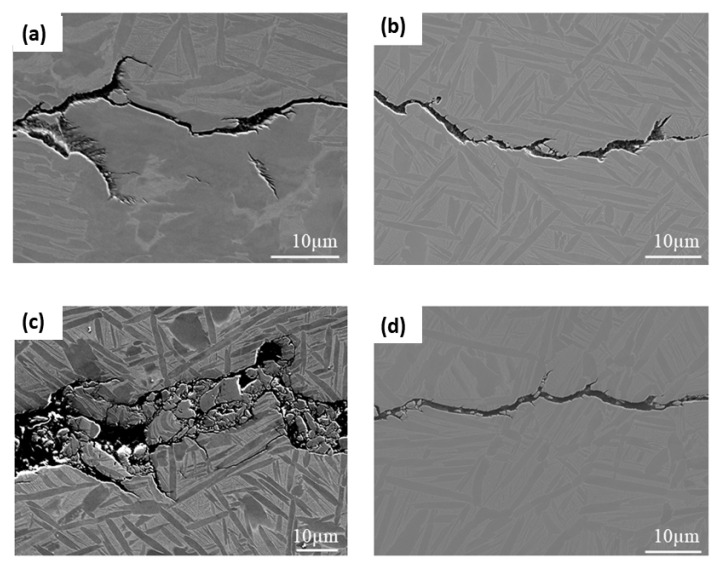
Micrographs (**a**) IP 200−500 °C (**b**) OP 200−500 °C (**c**) IP 200−550 °C (**d**) OP 200−550 °C. (**a**,**c**) show more evidence of creep damage ahead of the crack tip.

## Data Availability

Not applicable.

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
