# Peer review of "Thermo-Mechanical Fatigue Crack Growth and Phase Angle Effects in Ti6246"

_materials, 2022, doi:10.3390/ma15186264_

Round 1
Reviewer 1 Report
Dear authors, please find attached some comments.
Best regards

Author Response
Please see comments in the attached file

Reviewer 2 Report
The subject of research is relevant, tests for titanium alloy were carried out methodically correctly. But there are comments on the part of the discussion of the results and the novelty of the conclusions.
· Summing up the section "Introduction" the authors say that -«The purpose of this research is therefore to investigate and understand the behaviour of Ti-6246 under TMF loading conditions». At the same time, in this section, the authors do not touch upon the analysis of the resistance of high-nickel alloys to high-temperature corrosion and their resistance to the development of creep. At the same time, when discussing the results obtained, the authors compare the damage of titanium alloys and high-nickel alloys. I think it's not correct.
· Controversial, in my opinion, is the position of the authors on explaining the mechanism of fatigue crack propagation. It is well known that a mixed failure mechanism is observed as a result of low-cycle fatigue. As a rule, the crack propagates along the body of the grain until it reaches the interfacial boundary. In this case, depending on the energy state of grain boundaries, mosaic blocks, their width, and orientation angles, a crack can develop along interfacial regions. This process is accelerated in the presence of precipitates of secondary phases along the grain boundaries.
· Conclusions on the work repeat the text when discussing the results. For example, conclusions No. 2, 3 state the obvious.
· Self-citation should be reduced.
· To improve the article I recommend authors, for example, to get acquainted with the works DOI: 10.1007/s11003-021-00488-4
Author Response

(The authors gave the same response as above.)

Reviewer 3 Report
Dear Editor: I would like to express my deep thanks for inviting me to review the manuscript ID: materials-1824386-peer-review-v1
Title: An analysis of thermo-mechanical fatigue crack growth in the titanium alloy Ti-6246
Authors: Jennie Palmer, Jonathan Jones, Mark Whittaker and Steve Williams
Comments:
Abstract
Need to rewrite the abstract part according to the results.
Please modify this sentence “Within the gas turbine engine, the high transient thermal stresses developed due to variations in power requirements during a typical flight cycle give rise to the phenomenon of thermo- 9 mechanical fatigue (TMF)”.
Introduction part:
Please explain aim and novelty of this work introduction part.
Titanium alloys have seen extensive use in the gas turbine engine for the past 70 years 18 due to the unique balance of properties that they offer
Replaced by,
“Titanium and its alloys have seen extensive use in biomedical and aerospace industries particular in the gas turbine engine for the past 70 years due to the unique balance of properties that they offer [1,2].
[1] A. Jawaida, C.H. Che-Harona, A. Abdullah’ Tool wear characteristics in turning of titanium alloy Ti-6246, Journal of Materials Processing Technology 92-93 (1999) 329-334
[2] A.K. Gain, L. Zhang, M.Z. Quadir, Composites matching the properties of human cortical bones: The design of porous titanium-zirconia (Ti-ZrO2) nanocomposites using polymethyl methacrylate powders,Materials Science and Engineering: A 662, (2016) 258-267
Materials and Methods
Explain in detail the starting Ti6246 alloy, for example manufacturing process and so on
Results and discussion:
(i) Figure 6 setup schematic diagram put in Materials and Methods section
(ii) Figure 7 and Figure 8 combine together.
(iii) Explain the crack propagation mechanism in detain in Figure 15.
(iv) Please provide SEM images in Figure 13 instead of Profilometry of TMFCG fracture surfaces:
Conclusion:
Please concise the conclusion according to your results
RECOMMENDATION
After reviewing the enclosed manuscript for “Materials”, the present manuscript contains some kinds of scientific analysis but it is mandatory required to modify according to the preceding remarks. So, the manuscript can be accepted for publication after major mandatory revisions have been made.
Author Response

(The authors gave the same response as above.)

Round 2
Reviewer 1 Report
Dear authors, please see the attached document.

Reviewer 2 Report
After corrections and additions, the quality of results interpretation has improved
Author Response
No further revisions required
Reviewer 3 Report
Authors addressed all the comments in the revise manuscript.
Author Response
No further revisions required